# Complex Networks Theory for Evaluating Scaling Laws and WDS Vulnerability for Potential Contamination Events

**Matteo Nicolini**

Polytechnic Department of Engineering and Architecture, University of Udine, 33100 Udine, Italy; matteo.nicolini@uniud.it; Tel.: +39-0432-558742

**Abstract:** In the last few years, water utilities have recognized the importance of evaluating the safety of water distribution networks from the potential risks of contamination, arising from failures or from intentional (targeted) or random attacks. Research literature has been recently focused on the optimal design of efficient detection systems, generally expressed as the problem of the optimal placement of monitoring sensors. In this paper, we introduce a methodology for calculating an index of vulnerability that represents the tendency of an injected contaminant to spread over the network. Epanet quality simulations are performed in order to determine the distribution function of the number of potentially contaminated nodes. The results show how such distribution is overall fitted by a stretched exponential law. The comparison with an auto-similar, tree-like network (described by a power-law) allows the determination of the Vulnerability Index, which quantifies how "far" the behavior of a given system deviates from pure scale-freeness. It is analytically calculated by a two-fold approximation of the stretched exponential and provides an alternative way of evaluating robustness against random water contamination. Different networks can then be directly compared, in order to assess and prioritize control measures and interventions.

**Keywords:** water safety plan; scale-free; degree distribution; source tracing

## 1. Introduction

Water distribution systems (WDSs) are strategic, infrastructural networks designed for providing potable water to consumers. However, ageing pipes and deteriorating components may often affect the level of service, sometimes also with negative effects on the quality of the resource. As a consequence, water utilities have to pursue a continuous process of monitoring, control and operational management, in order to guarantee the delivery of safe water even under occasional circumstances of high demand or adverse situations arising from climatic or accidental emergencies.

In the few last years, Water Safety Plans (WSPs) have been recognized a valid tool for a thorough identification of the various hazards and related impacts on a WDS [1,2], allowing water managers and public institutions to perform global risk assessment and to prioritize control measures and strategic interventions in order to enhance overall reliability [3,4]. In this context, many studies have focused on quantitative resilience and robustness evaluation (recent reviews are in [5,6]), but less on reliability analysis for water quality (see [7] for a review).

One problem still unsolved is represented by the optimal design of monitoring systems for the prompt identification of random or targeted contamination events, which are, today, more reliable due to the rapid improvements achieved in sensor technology and data transfer and communications. Both single-objective and multi-objective optimization problems have been addressed [8].

In the last two decades, Complex Network Theory (CNT) has received increasing attention for the comprehension of a wide spectrum of real systems, ranging from physical infrastructures to social

communities [9,10]. In particular, the application of CNT to the design and operation of WDSs has attracted a growing number of researchers, because of its inherent capability of unveiling hidden properties not grasped by traditional analyses or modelling approaches [11].

CNT has been adopted for evaluating the topological characteristics and the resilience of a system [12], especially for expansion strategies [13]. Several authors investigated vulnerability-related issues, like node vulnerability under cascading failures [14–16], spectral methods to establish vulnerability areas [17,18], or methods for evaluating robustness under random or intentional attacks [19]. Other studies focused on the analysis of the formation of isolated communities [20] and on the segmentation of WDNs for the identification of District Metered Areas (DMAs) using general metrics and modularity [21,22]. The optimal sampling design has also been addressed with the modularity concept [23] and through a combination of classical optimization and CNT [24].

More recently, complex network theory has been adopted for the systematic classification of WDSs [25], and for the optimal design of a water distribution system, adopting a tradeoff between network cost and flow entropy as a measure of network reliability [26]. All such studies have been mainly focused on the topological aspects of WDSs. A step forward is taken in [27], where the standard definition of three centrality measures is extended in order to take into account the intrinsic relevance of vertices beyond the mere topology.

In this paper, we adopt a different approach, analyzing the functional relationship within a system through extended period simulations and the source tracing of a contaminant accidentally injected at junctions. In this regard, we recall that random or accidental contamination events can occur in real systems subjected to transients (which sometimes determine situations of negative pressure) or, more frequently, when localized rehabilitation work requires the temporary isolation and emptying of a certain number of pipes, thus sometimes facilitating the intrusion of undesired substances. More specifically, we introduce a methodology for evaluating a synthetic vulnerability index in order to compare different WDSs and their robustness with reference to the number of potentially contaminated nodes.

## 2. Materials and Methods

### 2.1. Complex Networks

Complex networks are ubiquitous in nature, society and technology [28], and can be represented by a set of multiple interconnected components (vertices or nodes) and a set of links (or edges). A fundamental distinction may be introduced according to the type of connection: technological (or material) networks are made by physical links, while virtual, conceptual and functional networks are based on several types of relationship between nodes (social networks or the World Wide Web are typical examples). Each network can be identified by an adjacency matrix, from which many characteristics can be derived: degree distribution, centrality measures (degree, closeness and betweenness), spectral properties, metric and topological structure, and global invariants [29,30].

In particular, degree distribution is fundamental for obtaining useful insights into the structure of a network, and indeed, some seminal studies classified complex networks according to their degree distribution as random [31], small-world [32] and scale-free [33]. Random and small-world networks are characterized by Poisson or exponential distributions, while scale-free networks exhibit power-law degree distributions, resembling their self-similarity and the fact they are formed by many nodes with a small number of connections and very few nodes with many connections, called "hubs".

Traditionally, such classification has also been adopted for distinguishing complex networks according to their behavior arising as a consequence of random failures or targeted attacks, i.e., structure affects function [34]; random networks are less vulnerable to intentional attacks, while scale-free networks are less vulnerable to random failures.

## 2.2. Cayley Regular Graphs

In order to evaluate the Vulnerability Index, we adopt a reference system represented by a particular kind of regular and self-similar tree-like network, that is, a Cayley regular graph [29]; Figure 1 shows an example with a bifurcation ratio of three, $X = 3$ (five levels are represented). As the number of levels grows, the network resembles a fractal object [35]. It is an idealized case of a WDS with only one reservoir, supplying a self-similar tree-like network.

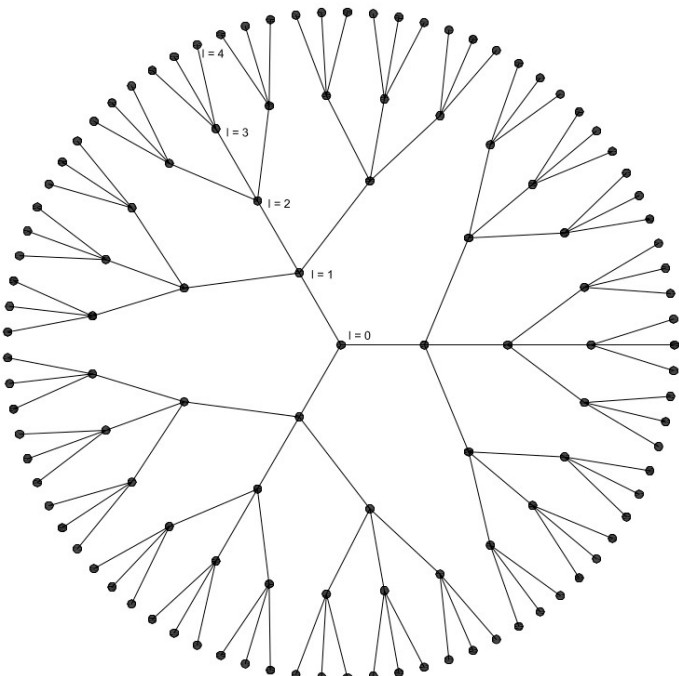

**Figure 1.** Example of a Cayley regular graph with five levels ($L = 5$) and a bifurcation ratio of $X = 3$ (solid lines represent actual links). The symbol $l$ represents the number of the level counted from the origin, so $l = 0, 1, 2, 3, 4$.

This represents a simple case in which the adjacency matrix of the WDS is the same as that of the virtual network, which we can obtain from quali-quantitative simulations, as described in the following section. In this case, the degree distribution of the number of nodes supplying a given number of "downstream" nodes is exactly power-law (of course, such a system is most vulnerable to targeted attacks, meaning that it can be completely compromised with one attack at its main "hub", or central supply node).

Some analytical expressions can be derived for such a simple system. The total number of nodes, $N_t(l)$, supplied by a given node that is at level $l$ is given by:

$$N_t(l) = \frac{X^{L-l} - 1}{X - 1} - 1 \equiv \frac{X^{l'+1} - 1}{X - 1} - 1 \tag{1}$$

where the symbol $l$ represents the number of the level counted from the origin of the graph (as shown in Figure 1), so $l = 0, 1, 2, 3, 4$, while $l'$ is given by: $l' = L - l - 1$, and can be thought as a "reverse" coordinate with respect to $l$. $N_t(l)$ can also be expressed as:

$$N_t(l) = \frac{X\left(X^{L-l-1} - 1\right)}{X - 1} \tag{2}$$

and in terms of $l'$, the following expression holds:

$$N_t(l') = \frac{X(X^{l'} - 1)}{X - 1} \tag{3}$$

from which it is clearly seen that $N_t(l')$ behaves as a power law:

$$N_t(l') \propto X^{l'} \tag{4}$$

The number of nodes at level $l$, $N_S(l)$, is given by:

$$N_S(l) = X^{L-l'-1} = X^l \tag{5}$$

while the total number of nodes in the graph can be calculated as:

$$N_{tot} = \sum_{l=0}^{L-1} X^l = \frac{X^L - 1}{X - 1} \tag{6}$$

The number of nodes at level $l$, dimensionalized with respect to the total number of nodes in the network, is then expressed as:

$$N_{S,a}(l) = \frac{N_S(l)}{N_{tot}} = \frac{X^l}{X^L - 1}(X - 1) \tag{7}$$

which can be written in terms of $l'$:

$$N_{S,a}(l') = \frac{N_S(l')}{N_{tot}} = \frac{X^{L-l'-1}(X - 1)}{X^L - 1} \tag{8}$$

We can easily obtain the cumulative value of $N_{S,a}(l')$, denoted $N_{S,a,cum}(l')$, as:

$$N_{S,a,cum}(l') = \frac{X^{L-l'} - 1}{X^L - 1} \tag{9}$$

which is approximated by the following power law:

$$N_{S,a,cum}(l') \propto X^{-l'} \tag{10}$$

Thus, by comparing Equation (4) with Equation (10), we can conclude that:

$$N_t(l') \cdot N_{S,a,cum}(l') \sim 1 \tag{11}$$

The variable $N_{S,a,cum}(l')$ represents the quantity we need in order to calculate the distribution function of potentially contaminated nodes. From Equation (11), it is easily seen that $N_{S,a,cum}(l')$ scales linearly with respect to $N_t(l')$ on a log-log plot (with a negative slope). In a real system, where loops are present and the direction of flows is not easily determined, we need to perform hydraulic simulations. The following section describes the methodology adopted for calculating the distribution function for a generic WDS.

### 2.3. Methodology for Determining the Virtual Network and Its Distribution Functions

The aim of the methodology is to assess the vulnerability of a WDS with respect to contamination threats. In other words, we qualify a system as highly vulnerable (high values of Vulnerability Index) if there is a high probability that big portions or even all of it could be potentially contaminated should a random attack occur (by random attack we mean an accidental injection of pollutant).

The key issue resides on the construction of a virtual network through quality simulations performed on the WDS, in such a way that we determine, for every junction, all the other "downstream"

nodes that are hydraulically connected to it. As we will see, such a virtual network is much more dense than that of the underlying WDS, the former being much more connected and complex than the latter, having a planar and almost regular structure. In this respect, we recall that WDSs are city spatial networks with topologies constrained by several environmental factors, and need to be robust with respect to random failures and intentional threats; for such reasons, WDSs are not generally characterized by scale-free features [36].

The analysis proceeds with the derivation of the out-degree distribution function of this virtual network, and analytical expressions are adopted for fitting the simulation results. In particular, the overall out-degree distribution is well approximated by a stretched exponential function, given by:

$$y = A \, e^{-Bx^C} \tag{12}$$

where *A*, *B* and *C* are positive constants. The vulnerability of a WDS may be estimated as the "distance", in a probabilistic sense, between the out-degree distribution of the corresponding virtual network and that of a "reference" system, which is, by definition, the least vulnerable with respect to random attacks. To this end, it is possible to measure such "distance" by integrating the difference between these two out-degree distribution functions. We will denote, by the *Vulnerability Index*, the measure of such difference.

Since Equation (12) in not analytically integrable, we opt for the separate approximation of the out-degree distribution between a power-law (for low-to-medium values of downstream supplied nodes) and an exponential function (for higher values). In this way, an analytical expression of the Vulnerability Index may be derived.

Moreover, we make the following assumptions: (1) contamination events originate only from one substance injection event at nodes; (2) the hydraulic conditions are not altered by the presence of contaminants (or by the volume of the injection); (3) each node may be the source of injection with equal probability (we therefore consider the likely intrusion of contaminants along pipes as if it were at junctions); (4) injection is modeled as steady-state, continuous and constant throughout the simulation; (5) pollutant decay or deposition processes (or even interactions with pipe walls) are neglected.

The whole procedure may be summarized in the following main steps:

1. Extended period simulations and source tracing analyses are performed with Epanet, one for each junction as a trace node, with sufficient simulation time in order to reach steady-state (or cyclic) conditions.
2. A directed functional network is built, where the adjacency matrix is characterized by having 1 at position $ij$ (for $i \neq j$) if, and only if, the contaminant injected at the $i$-th node reaches the $j$-th node, with a predefined concentration threshold (typically, 0.1%).
3. The out-degree distribution of this directed, complex network is what we have previously called $N_{S,a,cum}(l')$, and may be plotted on a log-log scale with respect to $N_t(l')$, the number of nodes supplied.
4. A stretched exponential function is fitted to the results of numerical simulations in such log-log plots. This allows the evaluation of its scaling behavior and can be compared to the power-law distribution describing the tree-like regular network. Accordingly, two separate fittings may also be obtained, namely, a power-law and an exponential function for subsequent analytical derivation.
5. The analytical formula derived in Section 2.4 allows an estimate of the vulnerability of the system. Thus, the more "distant" (in a probabilistic sense) the two distributions are, the more different the behavior of the real WDS is from a purely tree-like and auto-similar network, resulting in higher values of the index.

### 2.4. Analytical Expression of the Vulnerability Index: A Pragmatic Derivation

The relationship between $N_{S,a,cum}$ and $N_t$ for a generic WDN may be approximated by a stretched exponential distribution on a log-log plot, as emerged from the results (see Section 3). However,

in order to derive an analytical expression for the Vulnerability Index, the first part of the out-degree distribution is approximated by a power-law, and the second part by an exponential fit. If we denote $N_{inf} = 1$ and $N_{sup}$ as the lower and upper limits of the range for which power-law is adopted and $N_{sup}$ and $N_{max}$ as the corresponding limits for the exponential approximation, we can say that the following expressions hold ($x$ represents $N_t$):

$$N_{s,a,cum} = y = \begin{cases} \alpha\, x^{-\beta} \; if\; N_{inf} < x < N_{sup} \\ \gamma\, e^{-\delta\, x} \; if\; N_{sup} < x < N_{max} \end{cases} \tag{13}$$

On the other hand, the Cayley graph represented in Figure 1 is characterized by the following relationship, valid throughout the domain:

$$N_{s,a,cum} = y = 1 \cdot x^{-1} \tag{14}$$

In this way, the analytical expression giving the Vulnerability Index $V_I$—that is, the area between the out-degree distribution of the WDN and the tree-like Cayley graph—can be written as:

$$V_I = \int_{N_{inf}}^{N_{sup}} \left(\alpha\, x^{-\beta} - x^{-1}\right) dx + \int_{N_{sup}}^{N_{max}} \left(\gamma\, e^{-\delta\, x} - x^{-1}\right) dx \tag{15}$$

Keeping in mind that $N_{inf} = 1$, simple manipulation gives the following expression:

$$V_I = \frac{\alpha}{1-\beta}\left(N_{sup}^{1-\beta} - 1\right) + \frac{\gamma}{\delta}\left(e^{-\delta\, N_{sup}} - e^{-\delta\, N_{max}}\right) - \log_{10} N_{max} \tag{16}$$

## 3. Results

### 3.1. Characteristics of the Real WDSs Adopted for the Analyses

We analyzed four real networks in the north-eastern part of Italy. The layout of these systems is reported in Figure 2. G, T and U are WDSs supplying water to three medium-sized cities, while P WDS represents a system delivering water to twelve small towns. The main characteristics of these systems are reported in Table 1.

**Table 1.** Main characteristics of the real WDSs analyzed: $n$, number of junctions; $p$, number of pipes; $t$, number of tanks; $L$, total length of pipes; $Pop$, population served; $\rho$, network density; $<k>$, average node degree; $<\ell>$, average path length; $D$, network diameter; $<C_c>$, average clustering coefficient.

| WDS | $n$ | $p$ | $t$ | $L$ (km) | $Pop$ | $\rho$ | $<k>$ | $<\ell>$ | $D$ | $<C_c>$ |
|-----|-----|-----|-----|----------|-------|--------|-------|----------|-----|---------|
| G | 3958 | 4170 | 5 | 201 | 45,000 | 0.001 | 2.11 | 50.04 | 137 | <0.001 |
| T | 4140 | 4601 | 20 | 326 | 85,000 | 0.001 | 2.22 | 46.82 | 130 | 0.009 |
| U | 5772 | 6453 | 19 | 408 | 97,000 | <0.001 | 2.23 | 49.58 | 127 | 0.001 |
| P | 4619 | 5115 | 26 | 756 | 66,000 | <0.001 | 2.21 | 60.93 | 165 | 0.020 |

Each WDS has been simulated with the Epanet software, while model calibration has been performed with a genetic algorithm previously developed [37–39]. In particular, the demand pattern has been derived from SCADA (Supervisory Control and Data Acquisition) measurements and DMA monitoring data. Simulations are 48 h long, in order to be sure that the dynamics of pollutant diffusion are properly described by the simulated time horizon. We consider the contaminant reaching a given node if the concentration is above a certain threshold (in this case, 0.1% for an initial concentration of 100% at the point of injection). Table 2 reports the main properties of the virtual networks obtained from such simulations. It is important to note that the average node degree is very high, due to the way in which the virtual network has been defined (we recall that in real WDSs, the average degree

is very close to two). Moreover, the network density is an order of magnitude higher in the virtual networks than in the physical networks.

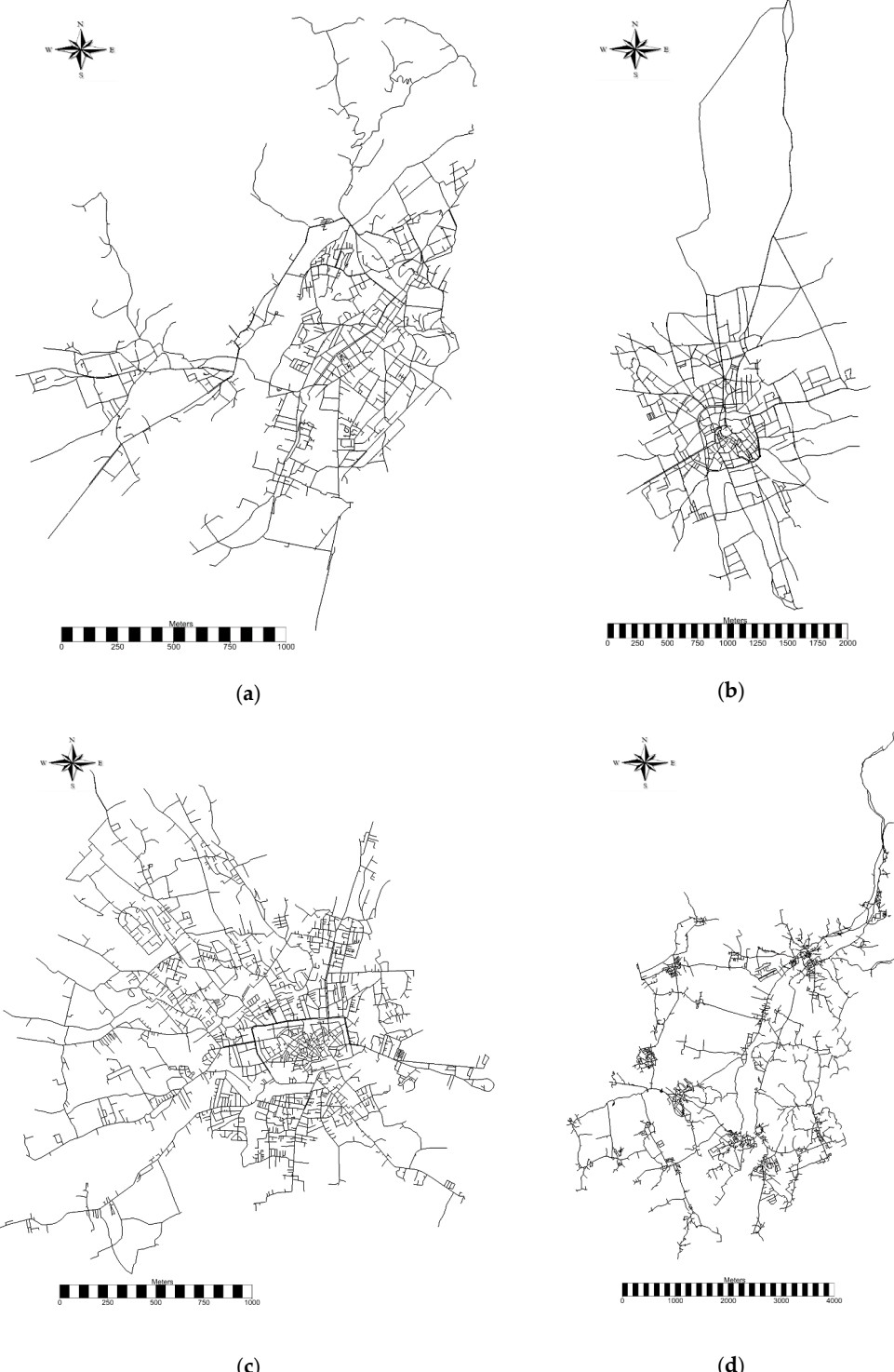

**Figure 2.** Layout of the four real systems adopted in the numerical analyses: (**a**) G water distribution system (WDS); (**b**) U WDS; (**c**) T WDS; (**d**) P WDS. The first three supply medium-sized cities, while P WDS supplies twelve small towns.

Some parameters in Table 2 have been calculated referring both to the directed and to the undirected virtual networks (symbols with the *u*-subscript); this last case is obtained by ideally removing the

directionality of the flow (and hence of the diffusion) of the contaminant. Although this is not realistic with respect to how real systems operate (contamination cannot actually flow upstream), this has been inserted only for comparison with the directed case.

**Table 2.** Main properties of the virtual networks built from extended period simulations and source tracing analyses (*n*, number of nodes; *m*, number of links). Symbols with the subscript *u* refer to the undirected network. $\rho$, network density; $<k_{out}>$, average out-node degree; $<\ell>$, average path length; *D*, network diameter; $<C_c>$, average clustering coefficient.

| WDS | $n$ | $m$ | $\rho$ | $\rho_u$ | $<k_{out}>$ | $<\ell>$ | $<\ell_u>$ | $D$ | $D_u$ | $<C_c>$ | $<C_{c,u}>$ |
|---|---|---|---|---|---|---|---|---|---|---|---|
| G | 3958 | 281,185 | 0.018 | 0.0359 | 71.042 | 1.1114 | 2.0970 | 5 | 8 | 0.204 | 0.443 |
| T | 4140 | 286,719 | 0.017 | 0.0335 | 69.256 | 1.5126 | 2.3657 | 8 | 13 | 0.184 | 0.433 |
| U | 5772 | 343,422 | 0.010 | 0.0206 | 59.498 | 2.0941 | 3.6016 | 11 | 10 | 0.289 | 0.583 |
| P | 4619 | 219,646 | 0.010 | 0.0206 | 47.553 | 1.6635 | 3.6291 | 5 | 14 | 0.270 | 0.578 |

Comparing such values with those in Table 1, we see that they are quite different, due to the way in which the tie relationships between nodes have been defined. In particular, the clustering coefficient values reflect the increased connectivity of such complex networks, as well the minor values assumed by average path length and network diameter, which suggest the emergent property of "small-worldness".

### 3.2. Scaling Laws of Different WDSs

Before applying the whole methodology described in the previous section, we first show the cumulative out-degree distribution in a log-log plot for several networks: Figure 3 reports the results of source tracing simulations performed on three literature networks, namely, KL [40], Exnet [41] and Wolf Cordera [42], and the four real systems previously described. The main objective of the paper is to focus on real systems, for which calibrated simulation models have been previously developed. The only purpose of inserting the results of literature networks is to also show that such systems may be, overall, described by the same exponential law. In Figure 3, exponential fit is also reported for each WDS, together with the correlation coefficient; although its values are quite high, a visual inspection shows that the fitting is much better in the right part of the plot, while for small-to-medium values of $N_t$, they are not properly represented by the exponential fitting.

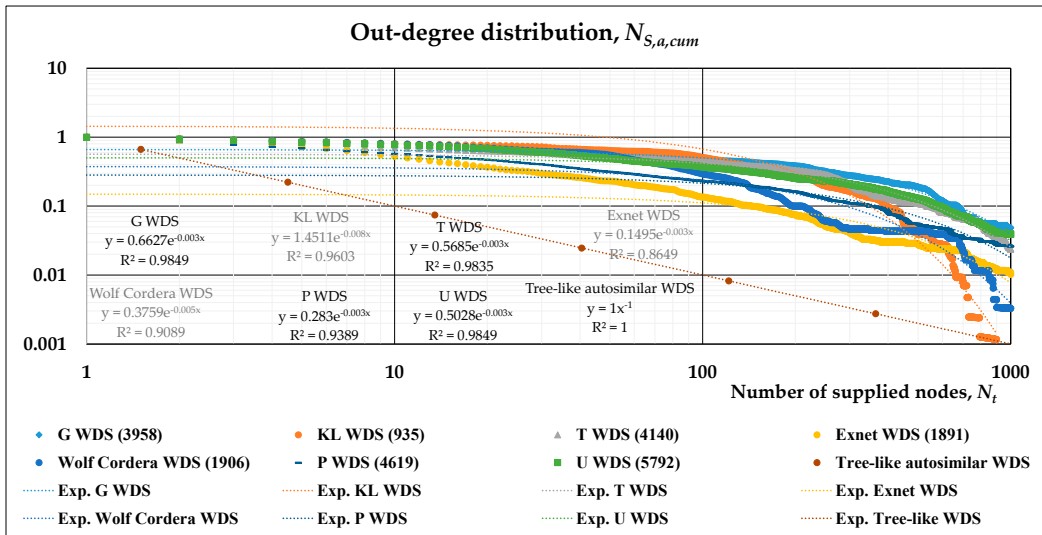

**Figure 3.** Log-log plot of cumulative out-degree distribution function for several WDSs, together with an overall exponential fitting. The tree-like auto-similar network of Figure 1 is also indicated, with its power-law distribution.

We can, anyway, assert that the behavior of the numerical results of different WDSs is quite similar on this log-log plot, confirmed by the fact that their distributions are located in a narrow range of the plot. This fact proves that there may be a "universal" trend in man-made WDSs arising from their topology, strongly connected to their growth, that is, their development (usually expansion) over time. Moreover, a closer look at the numerical results reveals that the first portion of the log-log plot is better approximated by a power-law distribution.

### 3.3. Scaling Laws Obtained by Different Fittings

Limiting our analyses to the four real systems, the "universal" trend may be described by approximating all the data with a stretched exponential distribution. Figures 4–7 show the results of such fitting, reported on different plots, since on a log-log scale, the *x*-axis is stretched differently. The coefficients of Equation (12) resulting from the overall approximation are reported in Table 3.

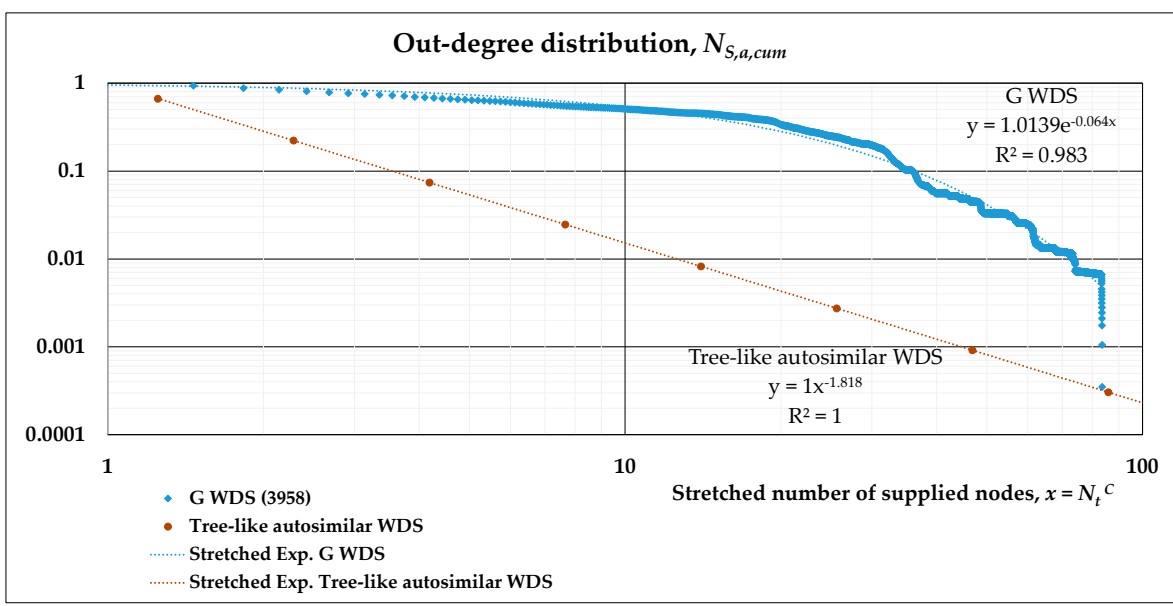

**Figure 4.** Cumulative out-degree distribution functions for G WDS, with stretched exponential fitting.

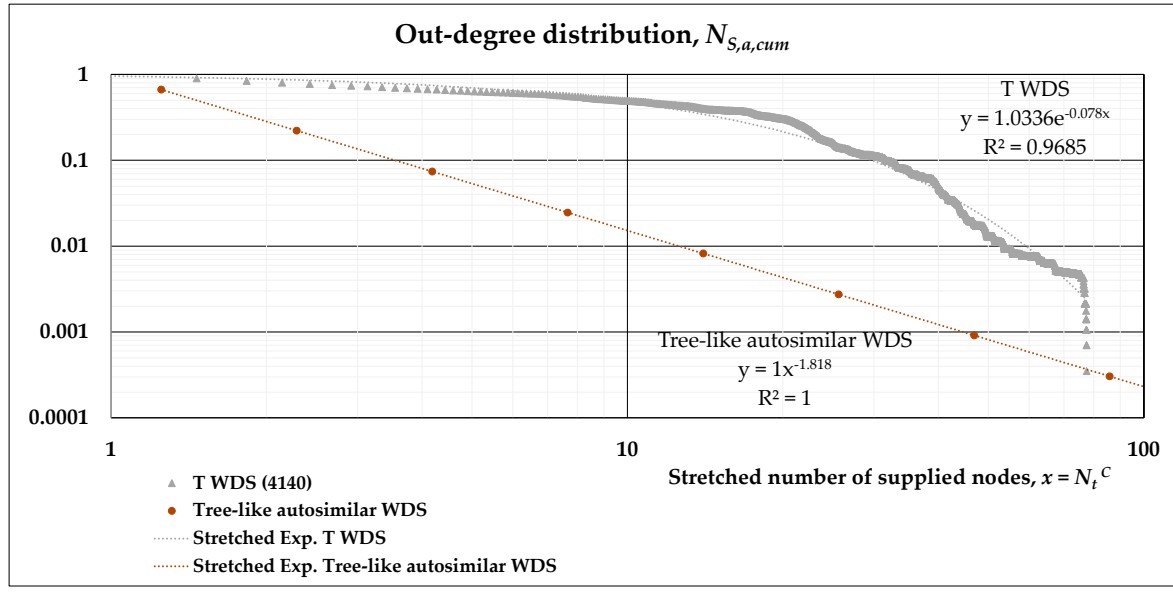

**Figure 5.** Cumulative out-degree distribution functions for T WDS, with stretched exponential fitting.

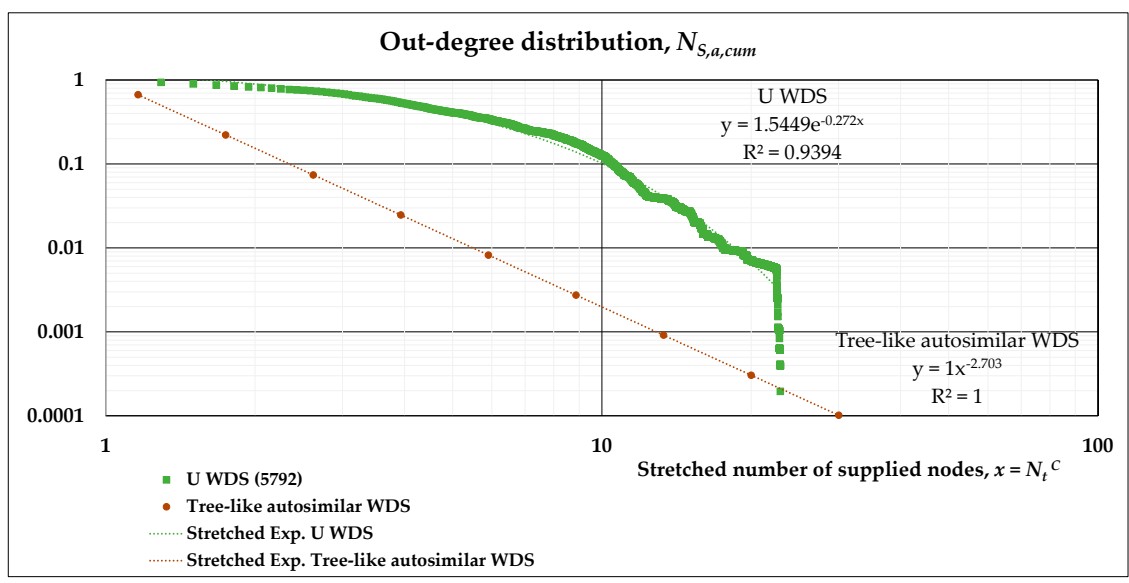

**Figure 6.** Cumulative out-degree distribution functions for U WDS, with stretched exponential fitting.

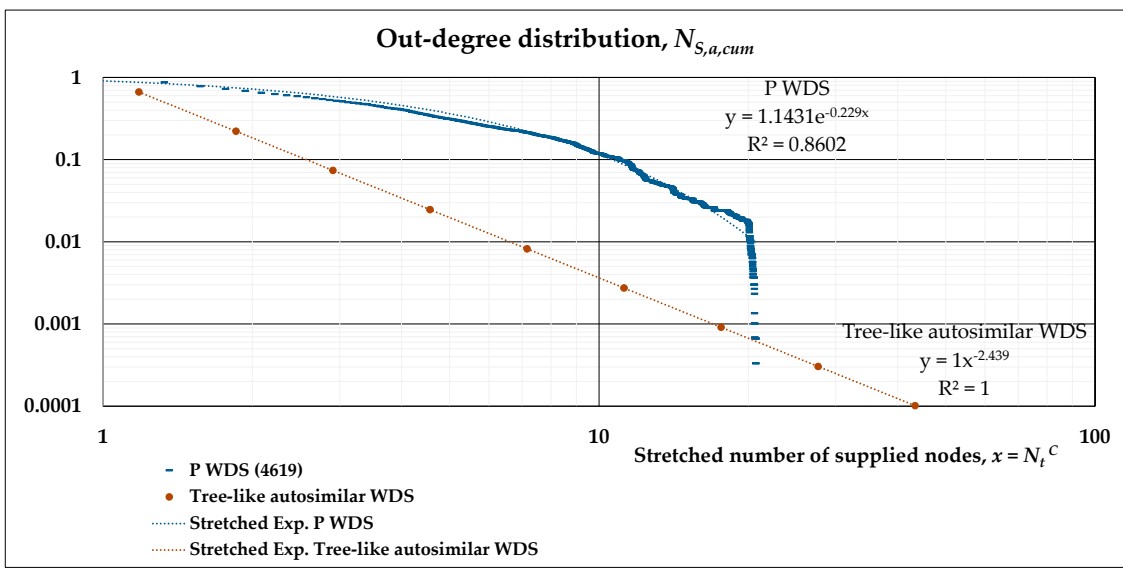

**Figure 7.** Cumulative out-degree distribution functions for P WDS, with stretched exponential fitting.

**Table 3.** Values of the coefficients obtained from the stretched exponential fitting.

| WDS | $A$ | $B$ | $C$ |
|:---:|:---:|:---:|:---:|
| G | 1.0139 | 0.064 | 0.550 |
| T | 1.0336 | 0.078 | 0.551 |
| U | 1.5449 | 0.272 | 0.370 |
| P | 1.1431 | 0.229 | 0.410 |

### 3.4. Determination of the Vulnerability Index for the Real Networks: A Pragmatic Method

With the aim of obtaining an analytical expression for the Vulnerability Index, $V_I$, we perform a separate approximation: power law (Figure 8) and exponential (Figure 9). Such curves allow the determination of, for each WDS, the coefficients, allowing the calculation of the $V_I$: The results are reported in Table 4. In order to calculate $V_I$, it has been assumed that $N_{sup} = 100$ and $N_{max} = 1000$.

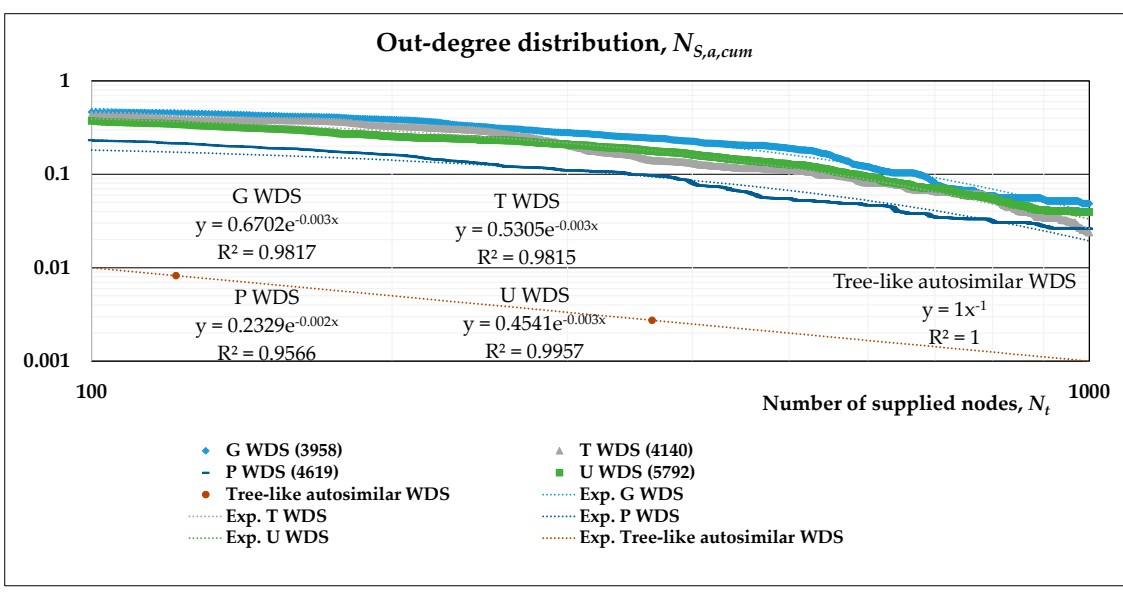

**Figure 8.** Cumulative out-degree distribution functions limited in the range 100–1000 $N_t$, together with their exponential fitting.

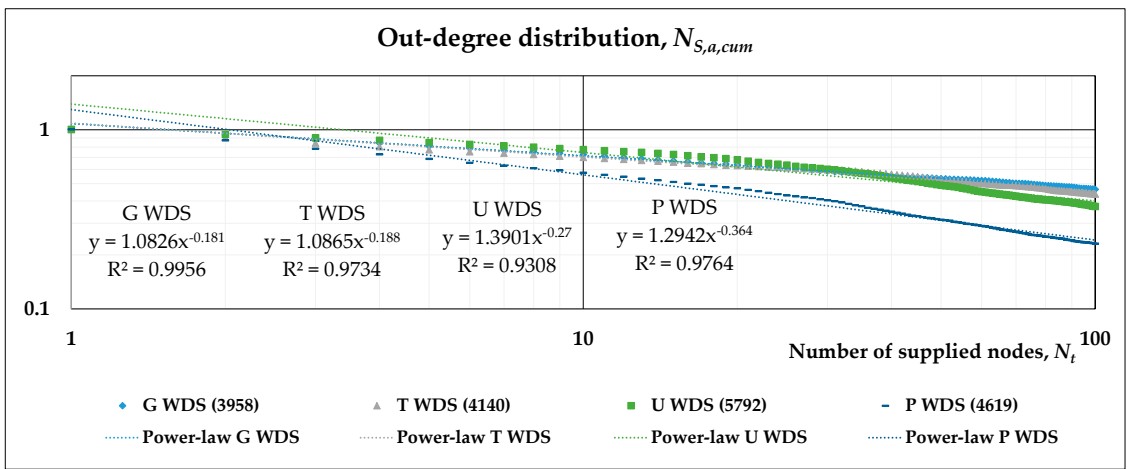

**Figure 9.** Cumulative out-degree distribution functions limited in the range 1–100 $N_t$, together with their power-law fitting.

**Table 4.** Values of power-law and exponential fitting coefficients, and resulting Vulnerability Index, $V_I$, for the real networks analyzed.

| WDS | $\alpha$ | $\beta$ | $\gamma$ | $\delta$ | $V_I$ |
|---|---|---|---|---|---|
| G | 1.0826 | 0.181 | 0.6627 | 0.003 | 205.73 |
| T | 1.0865 | 0.188 | 0.5685 | 0.003 | 182.91 |
| U | 1.3901 | 0.270 | 0.5028 | 0.003 | 165.83 |
| P | 1.2942 | 0.364 | 0.2830 | 0.003 | 98.22 |

## 4. Discussion and Concluding Remarks

The paper presents a new approach based on a synergy between Complex Network Theory and quali-quantitative Epanet simulations. The results obtained from source tracing analyses allow the determination of an adjacency matrix describing a virtual, directed network in which connections are established if, and only if, a substance injected at a given node reaches another one. In this way, a distribution function representing the potential contaminated nodes in a WDS can be derived, and its

distance (in a probabilistic sense) from a theoretical benchmark tree-like, auto-similar network may provide an overall assessment of its vulnerability to random contamination threats.

The paper introduces a vulnerability index useful for WDS operators from a practical point of view, since it summarizes with only one number the tendency of a WDS to be resilient with respect to random injections. According to this number, different systems can be directly compared, and WDS operators may be supported in planning interventions in their assets. This could also help decision makers and WDS operators in developing Water Safety Plans for their systems.

It is interesting to note that the diagrams in the preceding figures describe a scaling law which, starting at the peripheral portions of a network, "goes up" towards the main points of supply. In other words, on the left of the plots, the results show what happens at the terminal units of a WDS (small values of $N_t$), while when moving to the right of the diagram, we are virtually going "upstream" in the water distribution system, towards the reservoirs (which supply a large amount of $N_t$).

The analysis of the cumulative distribution functions tells us that at the peripheral portions of a WDS, the scaling law resembles that of a scale-free network, well described by a power-law distribution. However, the overall behavior of a real WDS is actually quite different from that of a scale-free network, being distant from the theoretical line representing the power-law distribution. This is in agreement with what was already observed by other authors [36,43], because of the requirement that real systems should be reliable with respect to random failures and intentional threats.

Such results may also be interpreted as a confirmation of the growth and evolution of such man-made systems, where after the initial planning and construction phases (typically regular and carefully designed), an expansion and aggregation of portions follow, in order to progressively connect the peripheral areas to the main system. In this way, while the main skeleton is almost regular in structure (exponential-like distribution), the successive portions are more tree-like and characterized by a more fractal-like behavior, reflected in a power-law out-degree distribution function.

This observation leads us to the possibility of quantifying the overall vulnerability of a given WDS by looking at the difference between its cumulative distribution function and the theoretical line representing a tree-like auto-similar network, the latter described by a power-law distribution.

The vulnerability index is a dimensionless number representing the distance between two probability distributions, one related to a real WDS and the other represented by an ideal, tree-like and auto-similar network with only one supply node (reservoir). The more distant the two distributions are, the more the behavior of the real WDS is different from that of the ideal network. Since the ideal network is highly resilient with respect to random contamination events, the higher the value of the vulnerability index (like WDS G), the less resilient the real WDS is to random events.

In order to derive an analytical expression, we have to split the overall approximation in two separate fittings: the power-law for low-to-medium values of $N_t$ and the exponential for higher values.

The results show that different systems can be compared and that a qualitative assessment of the vulnerability may be performed in order to prioritize interventions or for planning purposes when compiling water safety plans. The current limitation of the proposed approach resides in the assumption of the dynamics of contamination event, that is, the hypothesis that the injection lasts for a sufficient time until steady-state conditions are established all over the system. This is the very first step, however, but has the undoubted advantage of unveiling some basic mechanisms of WDS "functioning", as well as the tendency of some networks to be more prone to diffuse contamination in many areas. The next step will be that of relying on more realistic contamination events as described by transient injections.

One interesting issue to put into evidence is that it is possible to identify some critical portions of a WDS that may be affected by contamination events, not only for failures occurring nearby, but also for accidental injections coming from distant portions of the system. In this regard, it must be remembered that we are dealing with a virtual, directed complex network, and such particular zones could also serve as a further criterion in defining DMAs as being less vulnerable from the safety point of view. Research is currently being done in this direction.

**Funding:** This research received no external funding.

**Conflicts of Interest:** The author declares no conflict of interest.

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
