# Peer review of "Complex Networks Theory for Evaluating Scaling Laws and WDS Vulnerability for Potential Contamination Events"

_water, doi:10.3390/w12051296_

Round 1

Reviewer 1 Report

This an interesting paper developing a method to calculate the Vulnerability Index of a water distribution system to having contamination permeate a system from a point source intrusion event. The method analytically links a power law approximation of contamination spread with an exponential approximation.

The paper is quite unclear throughout the results and discussion sections. It is recommended that the authors expand upon both sections to more clearly identify:

  • How they are comparing their methodology to Epanet models.
  • What their results mean in terms of contamination of a WDS system and how they have defined vulnerability.
  • The limitations of their methodology.
  • How their methodology informs real world practice and decision-making.

Introduction

28: Suggest rewording to “may often affect” for readability

29: Perhaps the word is “pursue” instead of “purse”?

35: Remove “the” before “overall reliability”

71-74: Describing the organization of the paper is not necessary. Suggest moving the sentence

69-70 to previous paragraph and ending introduction there.

Materials and Methods

76-94: It is unusual to have background information presented in the methods section. However, I don’t mind it here as a reader because it is helping to provide background/framing to the methods adapted and trialed herein. It does make the paper read more like a textbook.

171-175: Can nodes be the sight of repeat injections of pollutants (repeat contamination events) in the model? Also, does the volume/amount of contaminant injection effect the model? Are contaminant events modeled as a static or dynamic injection? Does the contamination event occur at one time-step in the model or over a longer time length-sclae?

196: Please clarify what the boundary conditions mean (Nsup, Nmax)

Results

221: Could you please clarify the parameter, L. Is this the total length of all the pipes in the network or the maximum/median/minimum length of the network end-to-end?

221: Is rho the density of the nodes? Please clarify.

222: Are <l>, average path length, and D, network diameter, dimensionless?

222: Is <Cc> the average clustering coefficient of nodes?

238: The phrase “the first characteristic to put into evidence” is a bit awkward. Do you mean to say “the first parameter to note” or “It is important to note that the average node degree is very high”?

240: Suggest rewording: “Moreover, the network density is an order of magnitude higher in the virtual networks than the physical networks.” I think this is what you are saying?

242-245: Same clarifications as above

248: Could you comment in discussion on whether this is realistic to how systems operate? Is it likely to have contamination flow in a direction potentially opposite the water flow and contaminate upstream nodes? And provide more discussion on why you chose to allow this to be unconstrained and what value it provides to WDS operators.

267: Figure 3: The location of the exponential functions is confusing and the plot is too condensed to read in the y direction. Please place the labels for the x-axis and the title of the x-axis “Number of supplied nodes, Nt” below the plot, instead of along the line that corresponds to y=1. Perhaps you could put a table below the plot that includes the exponential functions and R2 values.

271-275: Also, there are results for KL WDS, Wolf Cordera WDS, Exnet WDS in Figure 3 that could be explained upon further in the results. What is the reader supposed to interpret from those three literature results compared to the model WDS systems you used?

273-274: Could you please clarify/expand on what you mean by this sentence, particularly related to WDS growth.

285-292: Same comments as Figure 3 for Figures 4-7. Possibly for these figures you could move the dot and dashed line legend below the plot and then do an inset legend defining the exponential equations and R2 values. Also, if you leave the x-axis labeled where it crosses y=1 I would recommend moving the labels above the axis so that the plotted data that trends downward does not overlap the labels.

299: Could you please explain a bit further the meaning of the Vulnerability Index values? Would WDS G be more vulnerable to contaminating more nodes from a single contamination event than the other WDS evaluated?

302-305: Same comments as above for Figures 8 and 9.

Could you please clarify throughout your results section, are the plotted points the results from the Epanet simulation?

Discussion

342-344: Can you expand on how this model could be put into practice and what WDS operators would gain from this model that they don’t currently have available to them when planning?

Can you please discuss the limitations to the model that you have proposed as well as the limitations on their use? Could you also discuss the next step in expanding the model or testing the model that could be done as confirmation.

Could you please discuss more about your results using a undirected vs. directed flow behavior and interpreting the impact it has on the vulnerability index?

Could you also discuss a bit more about the vulnerability index and how to interpret the results.

Reviewer 2 Report

No comments.

Well-written and thoroughly-documented research.

Author Response

No need to respond to Reviewer's comments.

Reviewer 3 Report

Congratulations for coming up with a scientifically sound, timely and well-organized research paper. I would highly recommend for its publication in the present form. 

Author Response

(The authors gave the same response as above.)

Round 2

Reviewer 1 Report

Thank you for adequately responding to the review.